# The Personality and Resilience of Competitive Athletes as BMW Drivers—Data from India, Latvia, Lithuania, Poland, Romania, Slovakia, and Spain

**DOI:** 10.3390/healthcare11060811

**Published:** 2023-03-09

**Authors:** Samir Rawat, Abhijit P. Deshpande, Radu Predoiu, Andrzej Piotrowski, Romualdas Malinauskas, Alexandra Predoiu, Zermena Vazne, Rafael Oliveira, Ryszard Makarowski, Karol Görner, Camelia Branet, Mihai Lucian Ciuntea, Doru Vasile Marineanu, Néstor Vicente-Salar, Davide de Gennaro

**Affiliations:** 1Faculty of Management Sciences, Symbiosis International (Deemed University), Lavale, Pune 412115, India; 2MBA (Sports Management) Program, Symbiosis School of Sports Sciences and Board of University Development, Symbiosis International (Deemed University), Lavale, Taluka Mulshi, Pune 412115, India; 3Faculty of Physical Education and Sport, National University of Physical Education and Sports, 060057 Bucharest, Romania; 4Institute of Psychology, University of Gdańsk, Jana Bażyńskiego 4 St., 80-309 Gdańsk, Poland; 5Department of Physical and Social Education, Lithuanian Sports University, Sporto g. 6, 44221 Kaunas, Lithuania; 6Department of Sport and Training Theory, Latvian Academy of Sport Education, Brivibas gatve 333, LV-1006 Riga, Latvia; 7Sports Science School of Rio Maior—Polytechnic Institute of Santarém, 2040-413 Rio Maior, Portugal; 8Research Centre in Sport Sciences, Health Sciences and Human Development, 5001-801 Vila Real, Portugal; 9Life Quality Research Centre, 2040-413 Rio Maior, Portugal; 10Faculty of Administration and Social Sciences, Academy of Applied Medical and Social Sciences in Elbląg, 82-300 Elblag, Poland; 11Department of Sports Education and Humanistics, Faculty of Sports, University of Presov, St. 17. novembra n., 08001 Presov, Slovakia; 12Faculty of Medical Engineering, Politehnica University of Bucharest, Splaiul Independentei 313, 060042 Bucharest, Romania; 13Faculty of Movement, Sport and Health Sciences, Vasile Alecsandri University of Bacău, Calea Mărășești 157, 600115 Bacău, Romania; 14Department of Psychology and Cognitive Sciences, Faculty of Psychology and Educational Sciences, University of Bucharest, 050663 Bucharest, Romania; 15Department of Applied Biology—Nutrition, Institute of Bioengineering, Miguel Hernández University (UMH), 03202 Elche, Spain; 16Institute for Health and Biomedical Research (ISABIAL), 03010 Alicante, Spain; 17Department of Management and Innovation Systems, University of Salerno, Via Giovanni Paolo II 132, 84084 Fisciano, Italy

**Keywords:** personality trials, Big Five, resilience, athletes, car drivers

## Abstract

Background: Individual differences in personality and resilience are related to a variety of social behaviors. The current study sought to answer the question of whether BMW drivers exhibit different personality profiles and resilience levels compared with drivers of other car brands. Participants and procedure: An international study was carried out in India, Latvia, Lithuania, Poland, Romania, Slovakia, and Spain on a sample of 448 athletes using the 20-item Mini-IPIP and the Resilience Scale. The results of BMW drivers (*n* = 91) were compared with the results of drivers of other German car brands (*n* = 357). Results: BMW drivers were characterized by higher neuroticism compared with drivers of other German car brands. They also showed higher resiliency, both in terms of total score and scores on the subscales of: personal coping competences and tolerance of negative emotions, tolerance of failures and perceiving life as a challenge, and optimistic attitude towards life and capacity for self-mobilization in difficult situations. The greatest difference was observed for the factor of tolerance of failures and perceiving life as a challenge. Using the Dwass-Steel-Critchlow-Fligner (DSCF) pairwise comparison test, gender differences between athletes (as BMW drivers and drivers of other German car brands, respectively) were discussed. Additionally, the results of the main logistic regression analyses emphasized that neuroticism represents a better predictor of BMW preference in the case of athletes (as drivers) than the scores obtained for resilience. Conclusions: BMW drivers differed from drivers of other German car brands only with regard to neuroticism. A higher level of neuroticism can affect mental health and the overall quality of life in athletes; aggression and distress management are essential. Athletes (as BMW drivers) also showed differences in resiliency levels. Understanding the mechanisms of behavior among BMW drivers is possible through considering their personality and individual differences.

## 1. Introduction

The Big Five personality factors (neuroticism, extraversion, openness to experience, agreeableness, and conscientiousness) continue to attract researchers’ attention. This is likely the result of the stability of this taxonomy, as well as of the model’s usefulness and effectiveness. The Big Five model’s taxonomy is consistently used in research of various occupational, language, and cultural groups worldwide [1]. The Big Five model is predictive of behaviors in professional [2] and personal settings [3]. Commonly accepted, the Big Five model allows researchers worldwide to systematically gather and compare data [4].

The traits comprising the Big Five model, which represent universal adaptive mechanisms, facilitate functioning in various social, occupational, physical, and cultural contexts [5], which is underscored by the evolutionary theory of personality [6]. Personality traits predispose individuals to certain behaviors and serve as adaptive mechanisms determining goal-oriented behaviors in various specific situations, including sport-related settings [7]. Individuals displaying traits which are desirable from the point of view of a given role will adapt and function more effectively in those roles and in their associated environments [8].

Personality allows for predicting both positive (e.g., health behaviors, [9] Doornenbal, 2021) as well as negative behaviors (e.g., bullying, [10]). Considering the sports domain, “personality profiles of karatekas specializing in kata or kumite are similar to the personality profiles of athletes from other sports. Namely: the personality profile of athletes is low neuroticism, high extraversion and conscientiousness” [11]. The specific sports branches practiced shape the personality traits in athletes (the personality features are dependent on experience); moreover, “personality and sport activities are interconnected in an ongoing process of two-way conditioning” [12]. It was found that emotional stability, openness to experience, extraversion, and conscientiousness are positively correlated with sports performance, while in the case of agreeableness, a negative link was reported [13]. Other researchers also highlighted that emotional stability and extraversion are associated with high conscientiousness and low neuroticism, respectively, in karate masters [14,15].

It is commonly believed that personality factors are important in the context of driving behaviors and are related to traffic accidents. Research has shown that certain personality traits, such as conscientiousness, neuroticism, and agreeableness, are strongly correlated with aggressive driving and road rage [16]. Dahlen and White point out that select personality traits (openness to experience, emotional stability, and agreeableness) may be useful in predicting dangerous driving behaviors [17]. Wei, Lee, Luo, and Lu have confirmed that among the personality traits, neuroticism is the most strongly related to risky driving behaviors: exceeding the speed limit, abnormal staying, and hard accelerating/decelerating [18]. Li established that drivers with high neuroticism and conscientiousness drive in a riskier manner, while individuals with high agreeableness are low-risk drivers [19]. Similar results were reported by Shen, Qu, Ge, Sun, and Zhang, who noted that neuroticism is negatively correlated with positive driving behaviors [20].

Driving safety is influenced by neurotic personality, which is related to risky and dangerous behaviors, as well as to an increase in the risk of traffic accidents [21]. Neurotic personality predicts dangerous driving behaviors [22]. High neuroticism leads drivers to become distracted and ignore traffic signs, including nonstandard ones, more often [23]. Drivers with high neuroticism also react more impulsively to traffic light changes [21]. Noticing slower reactions in other drivers may lead drivers with high neuroticism to aggressive behaviors. Drivers with high neuroticism generally display stronger stress reactions [24]. Meta-analyses showed that neuroticism and extraversion are related to dangerous behaviors, accidents, and injuries (employees of various professions were investigated) [25].

Cognitive abilities and personality factors function as independent predictors of driving behaviors [26]. Sutin et al. showed that neuroticism is related to lower memory performance, psychomotor performance, attention, executive function, and visuospatial abilities [27]. Emotional stability contributes to some cognitive abilities which are important for driving [28]. However, other individual differences are also related to driving behaviors. For instance, listening to rock music during driving increases speed variability and the frequency of changing lanes among individuals with choleric and sanguine personality profiles [29]. Another variable with relevance for driving behavior is resilience. Individuals with high resiliency cope better with stressful situations and situations which demand adaptation. Numerous studies highlight the role of resiliency in coping with difficult situations [30,31]. Resiliency increases psychological resources and allows for flexible adjustment to new life situations. Resiliency facilitates effective functioning in difficult situations as well as increased tolerance of negative emotions [32]. Individuals with high resiliency are able to better tolerate risks and experience lower levels of anxiety [33].

Currently, around twenty-three thousand lethal traffic accidents are reported in the European Union (EU) [34]. According to Tsai et al., “globally, more than 1.27 million people die in motor vehicle crashes each year, and 20–50 million people sustain injuries in vehicle crashes”, and the three major risk factors for fatal vehicle crashes are speeding, a history of benzodiazepine use, and involvement in motor vehicle crashes [35].

The majority of traffic accidents are avoidable contingent on the drivers adjusting their behaviors to traffic conditions and rules. Traffic accidents are chiefly the domain of men—women cause roughly one in four traffic accidents, and the risk of death for women drivers is over eight times lower than for men [36]. However, it should be noted that the number of men and women with drivers’ licenses is roughly equal. Cultural differences in traffic safety can also be observed. Countries with the highest number of lethal traffic accidents include Romania, Bulgaria, and Poland, while those with the lowest numbers are Sweden, Denmark, and Iceland [37]. For example, the number of victims per million inhabitants in Romania is double when compared with the European Union average [38].

In this sense, many insurance companies also require higher payments from drivers under the age of 26 [39]. This is because younger drivers are more often involved in accidents and drive in a more aggressive manner. Younger people are also more prone to risk-taking [40]. Insurance company data also tracks the car brands most often involved in traffic accidents. Not all brands are equally popular in all countries, leading to an over-representation of certain brands in the total number of cars, and, consequently, the total number of traffic accidents. Some studies showed that drivers of certain car brands are characterized by a less safe driving style than others. It is indicated that drivers of German brands (BMW, Audi, Opel), and BMW drivers in particular, drive less safely [41]. BMW drivers are stereotypically perceived as aggressive [42], driving faster than others [43], wild, and masculine [44]. Research showed that BMW drivers are more aggressive compared with other drivers [45]. It has also been noted that drivers of more expensive cars are more likely to drive in a more egotistic manner [46]. In the present study, the rationale behind choosing BMW vs. other brands refers to the driving style of BMW drivers; Ghayad [47], who summarized several studies, concluded that BMW drivers are the worst in terms of stopping for pedestrians, knowledge of traffic rules, and being the most disliked drivers on the roads (at a big distance, compared with drivers of other car brands). Moreover, BMW drivers were chosen for an exclusive comparison with other German car drivers because only BMW drivers have a social belief “to act in an aberrant (i.e., in abnormal) way” [45] (p. 89). Researches have suggested that BMW drivers act differently compared with other German cars “because they view their own presence on the road as a BMW-with-a-driver as qualitatively different to their presence as a non-BMW-with-a-driver, and therefore behave differently in their interactions with other road users” [45] (p. 89). Stereotypes were seen that poor driving is specific to drivers, “especially those who drive BMWs (although we have nothing against BMW)” [42] (p. 46).

In the current study, athletes were chosen rather than regular non-athlete drivers because professional athletes are often ‘macho-persons’ [48], “with ‘macho’ or narcissistic self-images” [49]. Not all athletes are champions, i.e., not all athletes have low levels of neuroticism [50], but neuroticism is the most strongly related to risky driving behaviors: exceeding the speed limit, abnormal staying, and hard accelerating/decelerating [18]. In addition, “macho-persons (i.e., athletes) not only act more aggressively on the road, but also they give preference to high-performance (for instance, often BMW) and sport-type cars and are more sensitive to image related issues” [45] (p. 90). Not least, athletes as drivers have not been the subject of much research so far. In the literature, we can find only a few studies on race car drivers [51,52]. They indicated the insufficiency of the research conducted so far in this field. For example, psychophysiological dynamics of race car drivers [53], the heart rate (HR) response in young male pilots in different conditions [54], or the physiological and thermal challenges in race car drivers were investigated [55]. Additionally, researchers were concerned with creating a mathematical model which can predict drivers’ performance (reaction time was measured) in a variety of driving conditions [56], and were interested in increasing the psychomotor qualities and, therefore, quality of life in senior drivers, through Visual-Motor Useful Field of View training [57].

Considering the previous information, the aim of the current study was to examine whether athletes as BMW drivers differ from drivers of other German car brands with regard to basic personality traits and resiliency. Because personality and resiliency are better predictors of behavior when considered together [58], both these variables were included in the study. To the best of our knowledge, the current study is the first to examine resiliency among athlete drivers.

Therefore, the following research questions were formulated:What are the resiliency levels among athlete drivers of various car brands (BMW and other brands)?What are the personality trait levels among athlete drivers of various car brands?Are there significant differences between BMW athlete drivers and athletes driving other German car brands with regard to basic personality traits and resiliency?Can we predict athletes’ preference for a car brand (based on personality traits or resiliency)?

## 2. Materials and Methods

### 2.1. Participants

A total of 448 participants from the following seven countries took part in the study: India, Latvia, Lithuania, Poland, Romania, Slovakia, and Spain. The participants were competitive athletes practicing different sports disciplines: track-and-field, martial arts (boxing, kickboxing, judo, fencing, karate, taekwondo), handball, soccer, aerobic and artistic gymnastics, swimming, volleyball, and basketball (fourteen sports disciplines in total). The inclusion criteria were: more than 18 years (seniors) and minimum five years of training in a specific sport discipline. Approximately 79% of the participants registered local/regional performances, while about 21% registered international/national performances, being considered experts/elite (with respect to the athletes’ standard of performance—Swann et al., 2015).

BMW athlete drivers (*n* = 91) were identified in the sample and were compared with athlete drivers of other German car brands (*n* = 357).

### 2.2. Measures

#### 2.2.1. Personality

Personality was measured with the 20-item *Mini-International Personality Item Pool* (Mini-IPIP), adapted by Topolewska et al. [59], which is a shortened version of the IPIP-Big-Five Factor Markers-50 (IPIP-BFM-50), measuring the Big Five personality traits according to Goldberg’s lexical model [60]. It consists of 20 items, arranged into 5 subscales (4 items for each subscale): extraversion (e.g., “I am the life of the party.”), agreeableness (e.g., “Sympathize with others’ feelings.”), conscientiousness (e.g., “Get chores done right away.”), neuroticism (e.g., “Have frequent mood swings.”), and intellect/imagination (e.g., “Have a vivid imagination.”). The participants indicated how accurate each phrase was for them, using a 5-point Likert-type scale (from “1”—very inaccurate to “5”—very accurate). In the current sample, reliability for the subscales, measured with Cronbach’s alpha, ranged from: 0.73—intellect/imagination to 0.85—extraversion (India); 0.78—conscientiousness to 0.88—neuroticism (Latvia); 0.79—intellect/imagination to 0.88—extraversion (Lithuania); 0.76—intellect/imagination to 0.86—neuroticism (Poland); 0.77—conscientiousness to 0.88—extraversion (Romania); 0.80—agreeableness to 0.88—conscientiousness (Slovakia); and 0.75—intellect/imagination to 0.85—extraversion (Spain).

#### 2.2.2. Resilience

To measure resilience, the *Scale for Measuring Resilience* (SPP-25) by Ogińska-Bulik and Juczyński was used [32]. It consists of 25 items, arranged into 5 subscales (5 items for each subscale): perseverance and determination in action (e.g., “If I have to do something, I usually do it right away.”), openness to new experiences and sense of humor (e.g., “I can look at situations from many different points of view.”), personal coping competences and tolerance of negative emotions (e.g., “ In difficult situations, I can cope with unpleasant feelings.”), tolerance of failures and perceiving life as a challenge (e.g., “I can draw conclusions for the future from my failures and mistakes.”), and optimistic attitude towards life and capacity for self-mobilization in difficult situations (e.g., “I always have an optimistic approach to life, regardless of the situation.”). The respondents indicated their agreement with each item on a 5-point Likert scale from “0”—I definitely disagree” to “4”—I definitely agree. The higher the partial scores on each scale or the total score, the higher the degree of resilience. In the current sample, reliability for the subscales, measured with Cronbach’s alpha, ranged from: 0.67—openness to new experiences and sense of humor to 0.78—personal coping competences and tolerance of negative emotions (India); 0.69—tolerance of failures and perceiving life as a challenge to 0.88—personal coping competences and tolerance of negative emotions (Latvia); 0.68—openness to new experiences and sense of humor to 0.75—optimistic attitude towards life and capacity for self-mobilization in difficult situations (Lithuania); 0.67—tolerance of failures and perceiving life as a challenge to 0.86—personal coping competences and tolerance of negative emotions (Poland); 0.68—perseverance and determination in action to 0.76—optimistic attitude towards life and capacity for self-mobilization in difficult situations (Romania); 0.70—openness to new experiences and sense of humor to 0.78—personal coping competences and tolerance of negative emotions (Slovakia); and 0.70—optimistic attitude towards life and capacity for self-mobilization in difficult situations to 0.76—perseverance and determination in action (Spain).

### 2.3. Procedure

The research was carried out between November 2021 and May 2022, with data collected (for the two questionnaires) through Google forms (Google LLC, Mountain View, CA, USA), as in previous investigations [61,62].

The snowball sampling technique was used as a recruitment technique to examine the participants: BMW athlete drivers and athletes driving another German car brand.

Regarding the 20-item Mini-IPIP questionnaire and the scale for measuring resilience (SPP-25), the final versions, in each language, were created through retroversion—a procedure used, also, in earlier studies [63].

### 2.4. Statistical Analysis

Mann–Whitney’s U test was used to see whether scores on the measured variables differed significantly. Mann–Whitney’s U test was chosen because it works fine with unequally sized samples. Condition, that the statistical power will diminish as the group sizes are unequal, has been taken into account. The calculated statistical power for unequally sized samples of the current study (a 1:4 ratio; total sample size: 448) is 0.83. G*Power tool to compute statistical power analyses was used [64]. The level of statistical significance was set at *p* < 0.05. The Dwass-Steel-Critchlow-Fligner (DSCF) pairwise comparison test was then used to examine the differences between four groups (male and female athletes as BMW drivers and male and female athletes driving other German car brands). The binomial logistic regression was carried out to predict athletes’ preference (based on neuroticism and resilience) for a car brand. IBM SPSS Statistics 27.0 software (IBM Corp, Released 2020, IBM SPSS Statistics for Windows, Version 27.0. Armonk, NY, USA) was used in the statistical analysis. Effect sizes for U-statistics were expressed as *r* (0.1 weak effect size, 0.3 moderate, 0.5 strong, ≥0.7 very strong [65]).

## 3. Results

Participants from the following countries participated in the study: India (*n* = 38; M_age_ = 30.81; SD = 12.84), Latvia (*n* = 72; M_age_ = 25.95; SD = 8.71), Lithuania (*n* = 56; M_age_ = 25.20; SD = 6.96), Poland (*n* = 121; M_age_ = 32.79; SD = 9.28), Romania, (*n* = 103; M_age_ = 24.52; SD = 5.43), Slovakia (*n* = 32; M_age_ = 25.83; SD = 8.83), and Spain (*n* = 26; M_age_ = 34.81; SD = 11.83). Considering the training experience in a specific sport discipline, the average in the entire sample was 11.6 years. Ninety-one BMW athlete drivers (50 males and 41 females) were identified in the sample and there were 357 athlete drivers of other German car brands (198 males and 159 females): Audi (36 males, 27 female athletes), Volkswagen (85 males, 68 female athletes), Opel (49 males, 35 female athletes), and Mercedes-Benz (28 males, 29 female athletes). The mean age of the BMW athlete drivers was 25.75 (SD = 8.32). The mean age of the athlete drivers of other German brands was 29.36 (SD = 11.19).

First, data were screened for missing values or outliers [66]. Using stem-and-leaf, no outliers were identified. Additionally, no missing values were detected (due to the online survey in which all questions had to be rated).

Table 1 shows the results of the median scores comparison and Mann–Whitney’s U test for BMW athlete drivers and athlete drivers of other German car brands.

The analyses show that BMW athlete drivers are more neurotic than athlete drivers of other German car brands. Effect size is r = 0.13, meaning a moderate to weak difference. Other personality traits (in terms of the 20-item Mini-IPIP scales) did not differentiate between these two groups. BMW athlete drivers also achieved statistically significantly higher total resiliency scores (r = 0.10—a weak effect size), as well as scores on the particular resiliency subscales. The subscale of *openness to new experiences and sense of humor* was also higher among BMW athlete drivers, although this difference did not reach statistical significance.

The resiliency results were roughly above 75 points for BMW athlete drivers (mean = 75.9, median = 76) and 72 points for the athlete drivers of other German car brands (mean = 72.9, median = 74). According to the norms, the mean score was 72.27; therefore, both of the results in the current study (at group level) are within the sixth sten which means an average level of resiliency.

In the next step, using the Dwass-Steel-Critchlow-Fligner (DSCF) pairwise comparison test, the existing differences between the four groups formed (male athletes BMW drivers, female athletes BMW drivers, male athletes driving other German car brands, and female athletes driving other German car brands, respectively) were examined, starting from the eleven dependent variables investigated (personality factors and resilience dimensions). Table 2 contains only the significant differences observed.

Statistical analysis of the data shows that male athletes as BMW drivers are significantly less agreeable than female athletes as both BMW or other German car brands drivers. Additionally, female athletes as BMW drivers are significantly more agreeable than male athletes who are drivers of other German car brands. Regarding neuroticism, it was found that male athletes as BMW drivers are significantly more neurotic than participants in all other three groups (the difference was marginally significant when compared with male athletes who are drivers of other German car brands). Additionally, when talking about resilience, male athletes as BMW drivers, as well as male athletes driving other German car brands, obtained a significantly higher score for personal coping competences and tolerance of negative emotions, optimistic attitude towards life and capacity for self–mobilization, and for the total Resilience score (only male athletes as BMW drivers in the case of the total Resilience result), compared with female athletes who were drivers of other German car brands.

Knowing that neuroticism and resilience are specific to competitive athletes as BMW drivers, we examined to what extent the two variables predict athletes’ preference for a car brand. To achieve this aim, binomial logistic regression was used, considering the defined categories—athletes as BMW drivers and athletes driving other German car brands. The data of the main logistic regression analysis are underlined in Table 3, predicting the likelihood of athletes preferring and driving a BMW car based on neuroticism and resilience.

The main logistic regression analysis (two separate regressions were performed) emphasized that both models are statistically significant (omnibus test model, *p* < 0.05). With respect to the Hosmer and Lemeshow goodness of fit test, *p* = 0.054 (for neuroticism) and *p* = 0.073 (resilience), pointing out that the models are not a poor fit. NR − χ^2^(1) = 12.28, *p* = 0.001, RES − χ^2^(1) = 4.12, and *p* = 0.042, meaning that the logistic regression models were significant. In the case of the competitive athletes, the scores for neuroticism represent a better predictor of BMW preference (as drivers) than those of the results obtained for resilience. The models correctly classified 79.7% (neuroticism) and 68.9% (resilience) of cases, respectively. Nagelkerke R^2^ (effect size index) shows a moderate to weak relation between neuroticism and athletes’ preference for BMW, and a weak relation between resilience and athletes’ preference (for driving a BMW).

It was asserted that higher levels of neuroticism and resilience are associated with an increased likelihood of driving a BMW car in the case of athletes.

## 4. Discussion

The human factor is responsible for the majority of traffic accidents. Excessive speed, not respecting right of way, and not adjusting speed for the weather conditions are responsible for over 95% of all traffic accidents [67]. The car’s condition, the weather, and the road infrastructure are responsible for a very small proportion of all traffic accidents, and it would be difficult to eliminate these factors entirely. Insurance company data and media reports show that traffic accidents are most often caused by young men. Moreover, there is a common belief that BMW drivers are aggressive. Thus, the current study involved a sample of competitive athletes and divided them into subsamples based on the brand of car they drive.

The aim of the current study was to estimate the personality profile and resiliency of BMW athlete drivers compared with athlete drivers of other car brands. Based on the literature [18], we expected that personality traits may be related to driving BMWs. BMW athlete drivers were characterized by higher neuroticism compared with athlete drivers of other German car brands. BMW athlete drivers’ neuroticism levels significantly exceeded the mean reported in the original IPIP-BFM-20 study [60]. Individuals characterized by higher neuroticism are more prone to reacting with frustration and anger at everyday events [68]. Individuals with high neuroticism do not cope well with stressful situations and have a tendency towards interpreting regular situations as threatening. In everyday situations, they more often interpret random occurrences as directed personally against them. They also show significant difficulties in controlling their own behavior [69]. Individuals with high neuroticism also engage in more reckless risk-taking behaviors [70].

As drivers, individuals with high neuroticism are characterized by a dangerous driving style, including purposefully breaking the speed limit, being aggressive towards other drivers [71], and using cellphones while driving [72]. Neuroticism seems to significantly impact driving behaviors from the point of view of causing dangerous situations, both for the drivers themselves as well as for other people. Neuroticism also plays an important part in shaping loyalty towards brands perceived as exciting [73]. Driving BMW cars can be seen as a source of excitement, partially due to the technical capabilities of the cars themselves. However, it remains an open question whether individuals with high neuroticism choose BMW cars or whether owning BMW cars increases neuroticism.

Considering success in sports and taking earlier reports talking about the Big Five model [1], researchers emphasized that “probably lower neuroticism and higher extraversion, openness to experience, agreeableness and conscientiousness” are specific to sport masters when compared with less successful athletes [74]. In the current study, higher levels of neuroticism were highlighted among athletes who were BMW drivers when compared with athletes driving other German car brands; the groups consisted of both top athletes and athletes with more local experience.

Macho personality is also related to an aggressive driving style [75]. This personality type is composed of such traits as, among others, perceiving violence as masculine and danger as exciting, and the cult of hardiness leading to self-control [76]. Such behaviors are commonly ascribed to BMW drivers [42,45]. It is possible that the social perception of BMW drivers encourages the purchase and use of these cars by individuals whose personality profiles are close to the macho stereotype [75]. At the same time, it is interesting to examine whether BMW cars are a differentiating factor among drivers of sports cars. Brands, including car brands, may be a reflection of the consumers’ personality and identity [77]. According to the self-congruity theory, individuals purchase cars of those brands which best reflect their actual or desired personality [78]. It seems interesting to examine which personality factors determine the choice of the BMW brand. High horsepower, rapid acceleration, and technical solutions increasing driving safety allow for a more dynamic, but also riskier, driving style. It has been shown that drivers of high-powered cars engage in dangerous driving behaviors more often [75]. High horsepower may characterize all cars in this brand, regardless of the model.

It has been widely documented in the literature that individuals with high resiliency are able to effectively cope with stressful situations [30]. To the best of our knowledge, this is the first study examining resiliency among athletes as drivers. The current results show that BMW athlete drivers are characterized by higher resiliency compared with athlete drivers of other German car brands. They achieved both higher total resiliency scores and higher scores on the subscales of *personal coping competences and tolerance of negative emotions, tolerance of failures and perceiving life as a challenge*, *optimistic attitude towards life, and capacity for self-mobilization in difficult situations.* The results seem to be in line with the observation that BMW drivers cope better with difficult traffic situations. However, BMW drivers are characterized by a more aggressive and less socially accepted driving style [45]. On the other hand, it has to be noted that both BMW athlete drivers and athlete drivers of other German car brands in the current study achieved average resiliency scores despite significant differences in some resiliency indicators.

The obtained results are important in the context of sports psychology. The human personality is visible in everyday behavior, consumer decisions, and the type of preferred music [8,79,80]. Sports coaches and psychologists need to pay attention to many aspects of athlete behavior. The brand of the selected car may indirectly indicate what personality traits a given athlete has. This is especially important in the context of Perepjolkina’s [45] (2009) research, which indicated greater aggressiveness in BMW drivers. Generally, athletes are closely monitored by their coaches, who must evaluate their behavior as a whole. The present research shows how car brand preferences are related to the sowing of basic personality traits and resilience. Resilience and controlled instrumental aggression are desirable and applauded in performance sport if they are within the boundaries of the game [81], and personality plays an important role in achieving high sports results [11]. However, sometimes “the fierce struggle for winning in competition and the *win-at-all-costs philosophy*” [82] can generate violent behaviors, mostly in the case of young athletes [83]. If, in this context, it can be added to a higher score for neuroticism—athletes are less emotionally stable and experience more negative effects, including irritability, anger, or anxiety [63]—the negative impact of such situations on athletes’ mental health and overall quality of life is ensured.

Thereafter, the present research can contribute to a better understanding of the functioning of athletes in various fields of their activity. The findings can be helpful for coaches and sports psychologists as they pay attention to the non-sport decisions and attitudes of athletes, which may affect their sports results. Consumer choices dictated by personality profile provide additional information to coaches and sports psychologists and allow them to better understand possible patterns of their behavior. The main reason why the authors believe that this study will be useful to the community is based on the belief that special attention should be paid to athletes who choose to drive a BMW car, given the lack of stress and aggression management. Based on the findings of this study, a recommendation can be made for those athletes who choose to drive a BMW car to implement aggression management programs.

In order to reduce distress and increase emotional balance in athletes (developing a better ability to deal with unexpected situations or mistakes) members of the multidisciplinary team, and especially sports psychologists, can use: cognitive and behavioral strategies, which have proven their effectiveness [84]; analytical relaxation and autogenic training; positive self-talk (inner monologue) for self-confidence; self-monitoring of emotional reactions [85]; the 4Ds for dealing with distress—which unifies strategies and exercise with the aim of restoring wellbeing and improving emotion regulation [86]; stress inoculation training (SIT)—seen as an educational program to improve self-control [87]; or *internal* techniques, e.g., breathing and meditation, which reduce hostility and impulsivity in athletes [88]. To acquire healthy behaviors, athletes with a higher level of neuroticism (in collaboration with specialists) could attempt mindful activities—the experience of the body as trustworthy and safe [89], could learn appropriate conflict resolution strategies, and improve communication skills, which are verbal, nonverbal, and paraverbal. Additionally, eustress (positive stress) can be induced in more neurotic athletes by guiding them to get involved in pleasant activities that offer them satisfaction [90]. Additionally, “an integrative model of intervention against psychological (chronical and traumatic) stress” can be used by sports psychologists (working to increase athletes’ emotional stability, mental health, and overall quality of life) comprising the following domains: emotional, behavioral, and cognitive [91].

However, a higher score for resilience (in the case of BMW drivers) offers an advantage in performance sports as well as allowing athletes to have a better ability to bounce back from failure and overcome barriers on the road to great performance.

Taken together, the results on personality and resiliency show that from among the personality dimensions, only neuroticism is negatively related with resiliency—the higher the neuroticism, the lower the resiliency [58]. Similar results have been reported in the original study on the scale for measuring resilience [32]. In the case of athletes as BMW drivers, resiliency and neuroticism are both at higher levels when compared with the values obtained by athletes who are drivers of other German car brands.

The findings of the current research extended previous investigations and also addressed gaps in the specialized literature, examining a less approached sample and topic—athletes as BMW drivers in terms of resilience and the Big Five personality traits.

### Limitations and Future Directions

The current study was carried out in only a few countries. Thus, it is unclear whether the results would be replicated in other countries and, therefore, replicating the present research is warranted in other countries. In addition, the current study also did not measure how long the athletes were BMW or other brand drivers, which may also impact the results. Other variables potentially related to car brand choice were also not analyzed, such as socioeconomic status or the social perception of BMW drivers in a given country. In future studies, the economic status of the people who use these car brands should be asked, taking into account that the personality traits and resilience of people with high and low incomes may differ. Additionally, only self-reported measures were used, and the obtained results were declarative, the aspects of possible desirable responses being known [82]. Nonetheless, it is worth mentioning that the large number of studied athletes represents a strength of the current research. In further studies, data could be gained by combining questionnaires with behavioral observations. Participants could also be asked about their traffic citations, speeding tickets, and so forth, in the last 1–2 years. Detailed questions could concern the cars’ specific parameters, such as tuning or applying various characteristic decals or decorative elements. Moreover, it is unclear how different the results will be when athletes as BMW drivers are compared with athletes driving other car brands (not German car brands), as well as a potential comparison between BMW athlete drivers from different countries of origin. Then, different studies must be conducted considering non-athletes (having less/greater driving experience), only novices or only elite/ expert athletes (as BMW drivers), and only athletes from a specific sport branch. Finally, it is not ensured that the findings were not influenced by the participants’ expertise and were solely because of driving a certain brand of car, because the present study has a cross-sectional and ex-post facto design.

It seems interesting to analyze the relationship between the brand of car owned and the attitudes towards traffic rules and safety. These attitudes could concern such rules as speed limits, driving under the influence of alcohol, using baby seats, complying with mandatory technical check-ups, and so forth. Results on drivers of various car brands may be valuable when analyzed in conjunction with the contents of these brands’ advertisements.

Future studies should examine the possible dynamics of personality and individual traits during a period of owning a BMW car (or a certain model of BMW car). It is also worthwhile to answer the question of whether differences in other traits can be distinguished among drivers based on the brand of car they drive, such as sensation seeking, narcissism, self-efficacy, or sense of coherence [92], considering the instrumental risk, the stimulating risk [62], or when talking about different factors of aggression: foul play, go-ahead and assertiveness—Makarowski [93], verbal aggression, anger, hostility, and physical aggression [94]. It is possible that the social perception of BMW drivers attracts individuals with a specific personality profile. This question also requires further study.

## 5. Conclusions

The conclusions of the current research, carried out in seven countries, emphasized that athletes as BMW drivers manifest a higher neuroticism level compared with drivers of other German car brands. Specifically, male athletes as BMW drivers are significantly more neurotic than participants in the other three groups (female athletes as BMW drivers and male and female athletes driving other German car brands), and significantly less agreeable than female athletes as both BMW and other German car brands drivers.

Athletes (male and female) as BMW drivers also showed higher resiliency in terms of total score and results on the following subscales: personal coping competences and tolerance of negative emotions, tolerance of failures and perceiving life as a challenge, and optimistic attitude towards life and capacity for self-mobilization in difficult situations compared with athletes driving other German car brands. Additionally, in terms of the total Resilience score, only male athletes (as BMW drivers) registered a significantly higher value than female athletes who were drivers of other German car brands (after pairwise comparisons).

In the case of the competitive athletes, the scores for neuroticism represent a better predictor of BMW preference as drivers than those results obtained for resilience. Various techniques are presented to be performed by sports psychologists in collaboration with athletes as BMW drivers, especially with male athletes, in order to reduce distress, increase emotional balance, to help them acquire healthy behaviors, and increase overall quality of life.

## Figures and Tables

**Table 1 healthcare-11-00811-t001:** Mann–Whitney U test results for the effect of car brand used on investigated variables.

Variables	Drivers	Median	Mann–Whitney U	Z	*p*
Extraversion	BMW	3.50	14,541.50	−1.551	0.121
	Others	3.25			
Agreeableness	BMW	3.75	16,128.00	−0.106	0.916
	Others	3.75			
Conscientiousness	BMW	3.75	16,005.50	−0.217	0.828
	Others	3.75			
Neuroticism	BMW	3.00	13,272.00	−2.710	0.007
	Others	3.00			
Intellect/Imagination	BMW	3.75	16,177.00	−0.061	0.952
	Others	3.75			
Perseverance and determination in action	BMW	15.00	14,962.50	−1.170	0.242
	Others	15.00			
Openness to new experiences and sense of humor	BMW	16.00	15,704.00	−0.494	0.621
	Others	16.00			
Personal coping competences and tolerance of negative emotions	BMW	15.00	14,058.00	−1.998	0.046
	Others	15.00			
Tolerance of failures and perceiving life as a challenge	BMW	16.00	13,732.00	−2.297	0.022
	Others	15.00			
Optimistic attitude towards life and capacity for self-mobilization in difficult situations	BMW	15.00	14,064.50	−1.989	0.047
	Others	14.00			
Total Resilience score	BMW	76.00	14,086.50	−1.957	0.050
	Others	74.00			

**Table 2 healthcare-11-00811-t002:** DSCF test results for pairwise comparisons with significant differences.

Group (Median)		W	*p*
		Agreeableness	
1 (3.50)	2 (4.25)	7.13	<0.001
1(3.50)	4 (4)	6.44	<0.001
2 (4.25)	3 (3.50)	−7.44	<0.001
3 (3.50)	4 (4)	8.42	<0.001
		Conscientiousness	
3 (3.50)	4 (3.75)	6.613	<0.001
		Neuroticism	
1 (3.50)	2 (2.75)	−5.70	<0.001
1 (3.50)	3 (3.25)	−3.54	0.059
1 (3.50)	4 (2.75)	−8.73	<0.001
2 (2.75)	3 (3.25)	4.79	0.004
3 (3.25)	4 (2.75)	−9.22	<0.001
Personal coping competences and tolerance of negative emotions			
1 (16)	4 (15)	−4.68	0.005
3 (15.5)	4 (15)	−3.73	0.042
Optimistic attitude towards life and capacity for self-mobilization in difficult situations			
1 (15)	4 (13)	−5.17	0.001
3 (14)	4 (13)	−4.56	0.007
		Total Resilience score	
1 (77.5)	4 (73)	−3.69	0.045

Note. 1—Male athletes BMW drivers, 2—female athletes BMW drivers, 3—male athletes driving other German car brands, 4—female athletes driving other German car brands, W—Wilcoxon value.

**Table 3 healthcare-11-00811-t003:** Results of the binomial logistic regressions analysis.

	Neuroticism	Resilience
Omnibus test—model	0.001	0.042
Hosmer and Lemeshow test (chi-square, *p*-value)	14.46 (0.054)	12.97 (0.073)
Nagelkerke R^2^	0.053	0.018
Overall percentage (Predicted − Percentage correct)	79.700	68.9
Wald test	11.737	3.923
*B*	0.550	0.020
SE	0.160	0.010
*Odds ratio* values	1.733	1.076
Confidence interval for *Exp(B)*	1.265–2.373	1.021–1.118

## Data Availability

The data presented in this study are available on request from the corresponding author.

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
