# Peer review of "The Personality and Resilience of Competitive Athletes as BMW Drivers—Data from India, Latvia, Lithuania, Poland, Romania, Slovakia, and Spain"

_healthcare, 2023, doi:10.3390/healthcare11060811_

Round 1
Reviewer 1 Report
This is a novel project; however, several areas must be improved to become publishable.
The major concerns are the unclear rationales for the study design.
1) What was the rationale behind choosing BMW vs. other brands? The background in the introduction was not very convincing. Why BMW? This is not clearly stated or explained in the manuscript. To me, based on the methods and study design, I don’t see any reason that the authors can’t just simply compare all German cars? What makes BMW so special than other German cars?
2) Given the athletes that were participated in this study came with different and various sports expertise, how do you manage the confounding factors? How do you ensure the findings were not influenced by the participants' expertise but solely because of driving a certain brand of car?
3) What was the rationale behind choosing athletes rather than regular non-athlete drivers? What makes athlete special in this study design?
4) Line 156: this paragraph seems to provide rationale behind this manuscript; however, insufficiency of available research should not be the only reason to conduct research. What is the main reason that the authors believe this research will benefit the community? The authors need to elaborate this part more in details.
5) The introduction section covered variety of topics and concerns, which also made the statement slightly confusing and distracting. Please consider reorganizing and revising the introduction section to be concise.
Minor comments:
Drivers vs. athletes vs. athlete drivers? Please be consistent with the term that was used. The parentheses were confusing for the readers.
Line 83 and 86: please do not quote a sentence if that is not exactly what the original sentence from another article. Please either quote the original sentence or rewrite the sentence.
Line 110: this sentence is confusing. Please revise. Do you mean “slower” when you said “less rapid”?
Line 112-113: which population this meta-analysis article was focused on?
Line 115: relationship of what? If the cognitive abilities and personality factors have been shown to be predictors, isn’t being predictors already stronger than just claiming a relationship?
Line 129: please spell out all the abbreviations when they first appeared in the manuscript, e.g., EU.
Line 129-133: how does this paragraph related to the following paragraph or even the topic of this manuscript?
Line 150-151: please check the grammar.
Line 185: Do you mean “track-and-field” by “athletics”? Please be consistent because athletic is a person while the rest of the items following were all sports items.
Line 180-198: Please move the results and findings, such as mean age, SD, sexes, practice years, etc., to the Results section. They should not belong to the Methods section.
Line 202-203: please spell out IPIP and IPIP-BFM since this is the first time it appeared in the manuscript.
Line 232-234: Please avoid parentheses and “- (dash)“ in a sentence. If something needs to be explained, please use a complete sentence for it. This has been very confusing throughout the entire manuscript.
Line 236: was there any report showing the reliability between different language version of the SPP-25? If so, please add to the manuscript.
Line 237-239: Please elaborate this sentence.
Line 248: What is the reason that the p was set at p≤ 0.05, instead of typically p<0.05?
Line 250: the methods included a prediction for athletes’ preference. This was not mentioned anywhere earlier in the manuscript. Please provide the rationale behind this.
Table 1: the authors proposed to use Mann-Whitney U test in the methods, but in fact, Table 1 reported both Mann-Whitney U and also Wilcoxon W test. Please clarify the rationale of using Wilcoxon W test and add it to the methods.
Line: 267: “Other personality traits did not differentiate between these two groups.” This is not an accurate statement since there were many p<0.05 on Table 1. Please clarify this.
Table 2: Please re-organize Table 2 since it is very confusing to read.
Author Response
Reviewer 1
This is a novel project; however, several areas must be improved to become publishable.
The major concerns are the unclear rationales for the study design.
1) What was the rationale behind choosing BMW vs. other brands? The background in the introduction was not very convincing. Why BMW? This is not clearly stated or explained in the manuscript. To me, based on the methods and study design, I don’t see any reason that the authors can’t just simply compare all German cars? What makes BMW so special than other German cars?
A: We added the following text in the introduction,:
In the present study, the rationale behind choosing BMW vs. other brands refers to the driving style of BMW drivers, which according to Ghayad [47] who summarised several studies and concluded that BMW drivers are the worst, in terms of stopping for pedestrians, knowledge of traffic rules, being the most disliked drivers on the roads (at a big distance comparing to drivers of other car brands). Moreover, BMW drivers were chosen for an exclusive comparison with other German car drivers because only BMW drivers have a social believe “to act in an aberrant (i.e., in abnormal) way" [45] (p. 89). Researches have suggested that BMWs drivers act differently compared to other German cars “because they view their own presence on the road as a BMW-with-a-driver as qualitatively different to their presence as a non-BMW-with-a-driver, and therefore behave differently in their interactions with other road users” [45] (p. 89). Stereotypes were seen that poor driving is specific to drivers “especially those who drive BMWs (although we have nothing against BMW)” [42] (p. 46).
2) Given the athletes that were participated in this study came with different and various sports expertise, how do you manage the confounding factors? How do you ensure the findings were not influenced by the participants' expertise but solely because of driving a certain brand of car?
A: Also, we added at Limits:
We can state as limitation that we do not ensure that the findings were not influenced by the participants' expertise but solely because of driving a certain brand of car because the present study has a cross-sectional and ex-post facto design.
3) What was the rationale behind choosing athletes rather than regular non-athlete drivers? What makes athlete special in this study design?
Athletes occupy a special place in our research. It is their functioning that interests us the most and we have devoted a lot of research, including international research, to this group. We try to study athletes from different perspectives in order to get to know this professional group as best as possible and through our research contribute to improving their results. E.g.
http://dx.doi.org/10.3390/healthcare10091770
https://doi.org/10.1186/s12887-023-03869-7
https://archbudo.com/view/abstract/id/13812
https://archbudo.com/view/abstract/id/14018
A: We added in the text:
In the current study athletes were chosen rather than regular non-athlete drivers because professional athletes often are ‘macho-persons’ [48], “with 'macho' or narcissistic self-images” [49]. Not all athletes are champions, i.e. not all athletes have low level of neuroticism [50], but neuroticism is the most strongly related to risky driving behaviors: exceeding the speed limit, abnormal staying, and hard accelerating/decelerating [18]. In addition, “macho-persons (i.e. athletes) not only act more aggressively on the road, but also they give preference to high-performance (for instance, often BMW) and sport-type cars and are more sensitive to image related issues” [45] (p. 90). Not least, athletes as drivers have not been the subject of much research so far. In the literature, we can find only a few studies on race car drivers [51,52].
[48] Stiles, D. A., Sebben, D. J., Gibbons, J. L., & Wiley, D. C. (1999). Why adolescent boys dream of becoming professional athletes. Psychological reports, 84(3_suppl), 1075-1085.
[49] Ferraro, T., & Rush, S. (2000). Why athletes resist sport psychology. Athletic Insight, 2(3), 9-14.
[50] Piepiora, P. (2021) Assessment of Personality Traits Influencing the Performance of Men in Team Sports in Terms of the Big Five. Front. Psychol. 12:679724. doi: 10.3389/fpsyg.2021.679724
4) Line 156: this paragraph seems to provide rationale behind this manuscript; however, insufficiency of available research should not be the only reason to conduct research. What is the main reason that the authors believe this research will benefit the community? The authors need to elaborate this part more in details.
A: We added at Discussion section:
Our research can contribute to a better understanding of the functioning of athletes in various fields of their activity. The findings can be helpful for coaches and sports psychologists, as they pay attention to non-sport decisions and attitudes of athletes, which may, however, affect their sports results. Consumer choices dictated by the personality profile provide additional information to coaches and sports psychologists and allow them to better understand possible patterns of their behavior. The main reason why the authors believe that this study will be useful to the community is that it is believed that special attention should be paid to athletes who choose and drive a BMW car, given the lack of stress and aggression management. Based on the findings of this study, a recommendation can be made to athletes who choose and drive a BMW car to implement aggression management programs.
5) The introduction section covered variety of topics and concerns, which also made the statement slightly confusing and distracting. Please consider reorganizing and revising the introduction section to be concise.
A: We revised and completed the Introduction section.
Minor comments:
Drivers vs. athletes vs. athlete drivers? Please be consistent with the term that was used. The parentheses were confusing for the readers.
A: Thank you for the observation, we revised the text and we consistently used - athlete drivers
Line 83 and 86: please do not quote a sentence if that is not exactly what the original sentence from another article. Please either quote the original sentence or rewrite the sentence.
A: Thank you for the observation, sentences in the text have been changed.
Line 110: this sentence is confusing. Please revise. Do you mean “slower” when you said “less rapid”?
A: Thank you again for the observation, sentence has been corrected.
Line 112-113: which population this meta-analysis article was focused on?
A: The meta-analysis was based on studies of employees of various professions - the phrase has been completed.
Line 115: relationship of what? If the cognitive abilities and personality factors have been shown to be predictors, isn’t being predictors already stronger than just claiming a relationship?
A: Thank you, sentence has been corrected.
Line 129: please spell out all the abbreviations when they first appeared in the manuscript, e.g., EU.
A: The text in the article has been supplemented.
Line 129-133: how does this paragraph related to the following paragraph or even the topic of this manuscript?
A: In this paragraph, we indicate that a major risk factor that causes fatal vehicle crashes in the European Union is excessive speed, which is associated with BMW car drivers. We added in the text:
- Ghayad (2023) [47] summarised several results of studies which concluded that BMW drivers are the worst, in terms of stopping for pedestrians, knowledge of traffic rules, being the most disliked drivers on the roads [...]
- neuroticism is the most strongly related to risky driving behaviors: exceeding the speed limit [...] [18]
- As drivers, individuals with high neuroticism are characterized by a dangerous driving style, including purposefully breaking the speed limit, being aggressive towards other drivers [67] [...]
Line 150-151: please check the grammar.
A: Thank you for the observation
Line 185: Do you mean “track-and-field” by “athletics”? Please be consistent because athletic is a person while the rest of the items following were all sports items.
A: We modified, thank you!
Line 180-198: Please move the results and findings, such as mean age, SD, sexes, practice years, etc., to the Results section. They should not belong to the Methods section.
A: The information has been moved to the results section.
Line 202-203: please spell out IPIP and IPIP-BFM since this is the first time it appeared in the manuscript.
A: The text has been supplemented.
Line 232-234: Please avoid parentheses and “- (dash)“ in a sentence. If something needs to be explained, please use a complete sentence for it. This has been very confusing throughout the entire manuscript.
A: The text has been corrected.
Line 236: was there any report showing the reliability between different language version of the SPP-25? If so, please add to the manuscript.
A: In the revised version, we presented the reliability, separately, for each country included in the study and for each tool used. However, since there are several countries, presenting all the values would make it difficult to read the article. Therefore, we decided to present the minimum and the maximum values and the personality dimensions, respectively the resilience subscales where these values were observed.
Line 237-239: Please elaborate this sentence.
A: The text has been revised and the phrase was moved to Participants section:
We mention that in each group (BMW athlete drivers, respectively athlete drivers of other German car brands) there were athletes from individual and team sports, from sports dis-ciplines with direct contact with the opponent, as well as from sports branches where there is no direct contact with the opponent (e.g., aerobic gymnastics, artistic gymnastics, swimming, volleyball, etc.), athletes who obtained local/regional results, and, also, having international/national sports performances.
Line 248: What is the reason that the p was set at p≤ 0.05, instead of typically p<0.05?
A: Thank you for the observation, p-value was set at p < 0.05.
Line 250: the methods included a prediction for athletes’ preference. This was not mentioned anywhere earlier in the manuscript. Please provide the rationale behind this.
A: Thank you, following this observation a fourth research question has been added:
Can we predict athletes’ preference for the car brand (based on personality traits or resiliency)?
Table 1: the authors proposed to use Mann-Whitney U test in the methods, but in fact, Table 1 reported both Mann-Whitney U and also Wilcoxon W test. Please clarify the rationale of using Wilcoxon W test and add it to the methods.
A: Thank you for the observation, we kept in Table 1 only the relevant information for the present study.
Line: 267: “Other personality traits did not differentiate between these two groups.” This is not an accurate statement since there were many p<0.05 on Table 1. Please clarify this.
A: Thank you for the observation, we completed the phase:
Other personality traits (in terms of the 20-Item Mini-IPIP scales) did not differentiate be-tween these two groups [...].
Table 2: Please re-organize Table 2 since it is very confusing to read.
A: Table 2 was reorganized.
Thank you for your valuable input to the study!
Authors

Reviewer 2 Report
Has the validity and reliability of the resilience scale been made for the people of these countries separately?
In this study, it may be that the validity and reliability of the scales in which only weak points were used were not suitable for people in different countries.
Again, the economic status of the people who use these car brands should be asked. The personality traits and resistance (resilience) characteristics of people with high and low incomes may differ. This issue should be mentioned at least in the study.
If there was a comparison between the people of each country, they should have been mentioned in the discussion.
Thanks
Author Response
Reviewer 2
Has the validity and reliability of the resilience scale been made for the people of these countries separately?
A: First, thank you for appreciating our work.
In the revised version, we presented the reliability for all countries together. Reliability in individual countries is at a similar level. However, since there are several countries and several psychological questionnaires, presenting all the values would make it difficult to read the article.
In this study, it may be that the validity and reliability of the scales in which only weak points were used were not suitable for people in different countries.
A: Thank you for the observation. In our study, we used tools that have already been used in many language versions.
https://ipip.ori.org/newItemTranslations.htm
Of course, the validity and reliability of these tools varies from country to country, but they assume acceptable values for international group research.
Again, the economic status of the people who use these car brands should be asked. The personality traits and resistance (resilience) characteristics of people with high and low incomes may differ. This issue should be mentioned at least in the study.
A: Thank you for the observation. We added at Limitations and future directions section:
In future studies, the economic status of the people who use these car brands should be asked, taking into account that the personality traits and resilience of people with high and low incomes may differ.
If there was a comparison between the people of each country, they should have been mentioned in the discussion.
A: Thank you for the observation. We added at Limitations and future directions section:
Also, it is unclear how different will be the results when athletes as BMW drivers will be compared with athletes driving other car brands (not German car brands), or a comparison will be carried out only between athletes as BMW drivers taking into account the country of origin.
Thanks
A: Thank you also for your appreciation.

Round 2
Reviewer 1 Report
Thank you for revising the manuscript. The manuscript requires minor text and grammar editing.